# Hesitant Fuzzy Entropy-Based Opportunistic Clustering and Data Fusion Algorithm for Heterogeneous Wireless Sensor Networks

**DOI:** 10.3390/s20030913

**Published:** 2020-02-08

**Authors:** Junaid Anees, Hao-Chun Zhang, Sobia Baig, Bachirou Guene Lougou, Thomas Gasim Robert Bona

**Affiliations:** 1School of Energy Science and Engineering, Harbin Institute of Technology, Harbin 150001, China; j.anees@hit.edu.cn (J.A.); 15bf02043@hit.edu.cn (B.G.L.); 2Satellite Control Facility (SCF-L) directorate, SE&T wing, Space & Upper Atmosphere Research Commission, Lahore 54000, Pakistan; 3Department of Electrical and Computer Engineering, Lahore Campus, COMSATS University Islamabad (CUI), Lahore 54000, Pakistan; drsobia@cuilahore.edu.pk; 4School of Life Science and Technology, Harbin Institute of Technology, Harbin 150001, China; thomasrobert@stu.hit.edu.cn

**Keywords:** hesitant fuzzy entropy, heterogeneous clustering, wireless sensor networks, opportunistic routing, multi-attribute decision modeling, data fusion

## Abstract

Limited energy resources of sensor nodes in Wireless Sensor Networks (WSNs) make energy consumption the most significant problem in practice. This paper proposes a novel, dynamic, self-organizing Hesitant Fuzzy Entropy-based Opportunistic Clustering and data fusion Scheme (HFECS) in order to overcome the energy consumption and network lifetime bottlenecks. The asynchronous working-sleeping cycle of sensor nodes could be exploited to make an opportunistic connection between sensor nodes in heterogeneous clustering. HFECS incorporates two levels of hierarchy in the network and energy heterogeneity is characterized using three levels of energy in sensor nodes. HFECS gathers local sensory data from sensor nodes and utilizes multi-attribute decision modeling and the entropy weight coefficient method for cluster formation and the cluster head election procedure. After cluster formation, HFECS uses the same techniques for performing data fusion at the first hierarchical level to reduce the redundant information flow from the first-second hierarchical levels, which can lead to an improvement in energy consumption, better utilization of bandwidth and extension of network lifetime. Our simulation results reveal that HFECS outperforms the existing benchmark schemes of heterogeneous clustering for larger network sizes in terms of half-life period, stability period, average residual energy, network lifetime, and packet delivery ratio.

## 1. Introduction

In WSNs, the wireless sensors are spatially distributed autonomous devices responsible for sensing the change in the required physical phenomena of their surrounding environment using a small microprocessor, a few transducers, a radio transceiver and a low-power battery [1,2]. These wireless sensor devices collaborate with each other for data sensing, data collection and aggregation purposes [2,3]. In order to reduce the communication overhead and to allocate resources to sensor nodes effectively, we need a topology architecture in which sensor nodes are organized in clusters. Each cluster includes one Cluster Head (CH) and several Cluster Members (CM) [4]. The multi-hop routing used in clustering topology to forward the sensed data from source to destination results in the overall decrease in energy consumption and also reduces the interference among sensor nodes due to specific allocation of timeslots for communication purposes [4,5]. The clustering topology in WSN can effectively optimize the data redundancy by significantly reducing the size of the collected data using data aggregation and data fusion techniques at CH level. The aggregated or fused data can then be forwarded to the Base Station (BS) for further processing and accurate decision making of interested events [4,5,6].

The recent literature shows that researchers have proposed a working–sleeping cycle strategy in WSN to save battery power in case the sensor nodes are idle and not performing any of the designated tasks. Alfayez et al. [7] discussed that the sensor nodes go to sleep to save their battery power and wake up before performing their routine operations. These node scheduling techniques can be categorized as synchronous and asynchronous working–sleeping scheduling. These node scheduling techniques are designed in accordance with the scenario to prolong the network lifetime and improve energy utilization by creating an opportunistic node connection between sensor nodes. According to [8,9,10,11,12], Opportunistic Routing (OR) is a paradigm for wireless networks that benefits from broadcast characteristics of a wireless medium by selecting multiple sensor nodes as candidate forwarders. In [10,11,12], a set of nodes were selected as potential forwarders that transmitted the data packet according to some special criteria after receiving data packet from their neighbors. This set of sensor nodes is called a Candidate Set (CS). The performance of OR significantly depends on several key factors, such as the OR metric, the candidate selection algorithm, and the candidate coordination method. The idea of OR can also be exploited in clustering topology of WSN to extend network lifetime, improve energy utilization, increase the packet delivery ratio (PDR) by adapting asynchronous working-sleeping cycle strategy.

Entropy in information theory is employed in various fields involving data analysis. Entropy uses the Probability Distribution Function (PDF) to statistically measure the degree of uncertainty in information sources [13]. The entropy H(X) of a random variable $X=\left\{x_{1},x_{2},\ldots x_{n}\right\}$ having probability distribution as p(X) can be given as $H\left(X\right)=-\underset{x\in X}{\sum }p\left(x\right)log_{2}p\left(x\right)$ for 0≤H(X)≤1 [14,15,16]. As we want to utilize the functionality of information entropy in clustering WSN, it should be kept in focus that CH or BS should not be hesitant or irresolute about any of their decisions regarding cluster formation and data fusion. Keeping in view the opportunistic connection between sensor nodes in heterogeneous clustering, we selected multiple parameters including an asynchronous working–sleeping cycle, status transition frequencies, residual energy, link quality factor in terms of signal-to-noise ratio, distance between sensor node and BS, and number of supported sensor nodes by a potential CH as our attributes of hesitant fuzzy set. Furthermore, we need Multi-Attribute Decision Modeling (MADM) to efficiently utilize our hesitant fuzzy set to generate hesitant fuzzy entropy matrix and determine our entropy weight coefficients [14,15,17,18].

In this paper, we propose a hesitant fuzzy entropy based opportunistic clustering and data fusion scheme in which a hesitant fuzzy set is created by acquiring the multi-attribute values of sensor nodes. Moreover, the hesitant entropy matrix is generated after finding entropy values of each attribute in the hesitant fuzzy set using data standardization process. Subsequently, the entropy weight model [14,15,17] is employed to determine the entropy weight coefficients for each sensor node and, finally, the threshold attribute values are determined and then compared with the original attribute values for making a decision about new CH. This entire process is part of the CH election procedure which is initiated by the BS and continued by every CH for all communication rounds. The BS creates a CH election set and adds sensor nodes in this set which have original attribute values greater than threshold attribute values. After that, the BS invokes these added sensor nodes by sending them a CH election set message. Furthermore, the hesitant fuzzy entropy technique is also used for reliable data fusion performed by CHs in such a way that CMs integrate and forward their sensed data to the assigned CH by removing the redundant information from the sensed data. CHs perform data aggregation on integrated data packets by concatenating them into a single larger data packet of a specified length. Later on, CHs exploit hesitant fuzzy entropy and entropy weight coefficients to detect the change in sensed data periodically and then send the aggregated data to the BS upon detecting that change in sensed data [19,20].

The rest of the paper is organized as follows: Section 2 describes the related research work conducted for heterogeneous clustering in WSNs, entropy-based clustering schemes, hesitant fuzzy entropy, and OR. System modeling is presented in Section 3. Our proposed scheme, HFECS, is presented in Section 4. Section 5 describes the case study of HFECS in neighborhood area networks. The performance evaluation and simulation results are described in Section 6. Finally, Section 7 and Section 8 discuss and conclude the paper, respectively, and provide some future research directions.

## 2. Related Research Work

Various researchers have focused on proposing different routing protocols for WSNs based on different parameters, such as end–end delay, successful packets delivered to sink, network lifetime, overall energy consumption, control packet overhead, and sink node mobility, etc. Ogundile et al. [21] presented a detailed survey for energy efficient and energy balanced routing protocols for WSNs. The authors argued that energy efficiency, packet delivery ratio and average end–end delay are the most critical and significant parameters for delay-tolerant and delay-sensitive applications involving WSNs. Furthermore, in this survey, the taxonomy of cluster-based routing protocols for WSNs was also discussed in terms of the energy efficiency and energy balancing aspects. Routing protocols in WSNs can be segmented into two main categories, i.e., hierarchical and non-hierarchical routing protocols. Non-hierarchical routing protocols are proposed on the basis of overhearing, flooding, and information related to the advertisement of the sink’s position through agent node selection, whereas hierarchical routing protocols are proposed on the basis of (i) grid-based, tree-based, cluster-based and area-based routing. Different hierarchical routing protocols have their own merits and demerits, but as far as cluster-based hierarchical routing protocols are concerned, researchers have been challenged with a task of achieving an optimal balance between end–end delay and energy consumption [4,5,6,21].

Yang et al. [22] introduced an emerging concept—the ‘utilization of working–sleeping cycle’ of sensor nodes to prolong the network lifetime. The working–sleeping cycle can be segmented into two categories—synchronous and asynchronous working–sleeping cycle. Recent studies have revealed that a synchronous working–sleeping cycle in sensor nodes could lead to an improvement in energy consumption, but significant contribution is required for efficient synchronization of sensor nodes. Ng et al. [23] presented an energy-efficient synchronization algorithm for sensor nodes in which counter-based and exponential smoothing algorithms could improve the energy consumption using adaptive adjustment of traffic and wakeup period. Moreover, the asynchronous working–sleeping cycle strategy was explored in recent studies as well. Asynchronous working-sleeping schedules depend on network connectivity requirements in terms of traffic coverage area [8,9]. Mukherjee et al. [24] proposed an asynchronous working-sleeping schedule technique in which required network coverage was achieved with a minimum number of awake sensor nodes. Due to the independent working-sleeping cycle of each sensor node in asynchronous working-sleeping strategy, opportunistic node connection can be established between sensor node and its neighbors, thus we need an Opportunistic Connection Random Graph (OCRG) theory to form a spanning tree formation of sensor nodes having opportunistic node connections between each other. In [25], Norman et al. proposed a novel random graph modeling for heterogeneous sensor networks based on different transmission ranges and a new routing metric supporting opportunistic node connections. Anees et al. [26] proposed an energy-efficient multi-disjoint path opportunistic node connection routing protocol for smart grids (SGs) neighborhood area networks (NAN) inspired by opportunistic connection random graph theory in WSNs with sink mobility. This routing protocol reduces the overall energy consumption, increases the PDR, increases the network lifetime and decreases the end–end network delay against existing benchmark schemes.

Many recent studies have addressed the problem of WSN clustering. Low Energy Adaptive Clustering Hierarchy (LEACH) variants were proposed by Liang et al. [27], Handy et al. [28] and Khediri et al. [29] in which each round of communication is segmented into two phases, i.e., (i) the setup phase, in which limited number of sensor nodes are selected as CH depending upon their probability values and CMs of that CH, (ii) each CH assigns a particular timeslot to every CM in order to avoid collision. Each CM forwards the sensed data only in that timeslot. Various researchers have also proposed many variants of LEACH in [28,29,30,31,32,33]. Smaragdakis et al. [31] proposed the heterogeneous aware Stable Election Protocol (SEP), which is based on weighted election probabilities of sensor nodes for selection of CH depending upon their residual energies. In SEP, the energy heterogeneity problem is characterized using node classification, in which nodes are either labeled as normal or advanced. This protocol ensures that, during CH election procedure, CH is selected randomly based on the fraction of energy of each sensor node, thus assuring a uniform energy usage of all sensor nodes. Foregoing this view, Femi et al. [34] presented an extension of SEP which is known as SEP-Enhanced or SEP-E. In SEP-E, the energy heterogeneity problem is characterized using a three-node classification, in which nodes are labeled as normal, intermediate and advanced sensor nodes. According to [34], the main goal of SEP-E is to achieve a robust self-configured heterogeneous WSN with a longer network lifetime of sensor nodes and better energy utilization. Sharma et al. [33] proposed a Heterogeneity-aware Energy efficient Clustering (HEC) protocol, which is based on three different phases and CH is selected in every phase in a different manner, i.e., (i) in the first phase, sensor nodes labeled as advanced are allowed to participate in CH selection, (ii) all sensor nodes, irrespective of their residual energies, are allowed to participate in CH selection procedure with equal probabilities, (iii) for the last phase, direct transmission to BS is preferred instead of clustering, due to low residual energy of sensor nodes.

Qing et al. [35] suggested a novel heterogeneous WSN supporting a Distributed Energy-Efficient Clustering (DEEC) scheme in which energy heterogeneity is characterized using normal and advanced sensor nodes. Additionally, Saini et al. [36] proposed an extension of the DEEC protocol which is known as the Enhanced Distributed Energy-Efficient Clustering (E-DEEC) scheme, in which the energy heterogeneity problem is addressed using three-node classification, i.e., normal, intermediate and advanced. In E-DEEC, the CH selection probabilities are not adapted as per the energy levels of sensor nodes. Javaid et al. [37] proposed a heterogeneous network model based on Enhanced Developed DEEC (ED-DEEC or DEEC-E) which is established on two-node power classification and three energy levels of sensor nodes for dynamically modifying the CH election probability. Manjeshwar et al. [38] introduced a novel energy-efficient protocol known as Threshold sensitive Energy-Efficient Sensor Network (TEEN) protocol for reactive networks. In TEEN, each CM in a cluster takes turns to become the CH for a time interval, called the cluster period. After every cluster period, the CH broadcasts a soft and hard threshold to its CMs along with other attributes. CMs only send the sensed data to its CH by changing the status from working to sleeping, only if the data values are in the range of interest, keeping in view the hard threshold. In addition to this, CMs send the data to CH by changing their status from sleeping to working only if their data values change by at least the soft threshold. Although there are no collisions between data transmissions due to TDMA scheduling, TEEN introduces an extra delay during the reporting of time-critical data.

It has been revealed through a detailed literature review that most of the clustering schemes consider attributes such as residual energy, distance to the BS, etc. as parameters of criteria. Meanwhile, the Entropy-Based Clustering Scheme (EBCS) also considers cluster load in terms of supported sensor nodes as a key parameter of criteria for the CH election procedure. [39]. The cluster load can be defined as the number of CMs that can be efficiently handled and supported by the current CH. EBCS includes remaining energy, distance to the BS, and the sum of distances to neighboring sensor nodes as other parameters of criteria besides cluster load. In EBCS, a new method is introduced to predict the residual energy at the start of the next round of communication based on consumed energy. This new method is used to select the CH for next round of communication. Energy heterogeneity is characterized using sensor nodes labeled as normal, intermediate and advanced in EBCS. In previous studies, several researchers have adopted entropy weight coefficient method for making decision in clustering environment [14,15,16,18]. Entropy weight-based multi-criteria decision routing is a routing technique in which the next hop decision is based on Multi-Criteria Decision Analysis (MCDA) and Multi-Attribute Decision Modeling (MADM) using entropy weight coefficients [14,15,17,18]. Qiang et al. [16] presented an optimization of objective and subjective weights based on fuzzy MADM routing for selection of next best hop in a heterogeneous WSN. Authors in [16] utilized the entropy weight coefficient method to avoid excessive deviation from the objective weights. Wang et al. [14] developed an index system for capacity assessment using entropy weight coefficient method.

Xia et al. [40] discussed hesitant fuzzy information aggregation in decision making problems. Xia et al. [41] discussed the hesitant fuzzy entropy, cross entropy, and their usage in MADM applications. Su et al. [42] merged the fuzzy logic method with clustering in order to propose a new data fusion method based on a fault tolerant WSN for improving the availability of communication bandwidth. In order to reduce the data processing load on BS and efficiently distinguish the authenticity of archived data, Izadi et al. [43] proposed a wireless sensor data fusion method based on fuzzy theory to improve the service quality in WSNs. Chaurasia et al. [44] presented an adaptive fuzzy logic algorithm to address the inaccuracies in data fusion. Zhai et al. [45] developed Hesitant Language Preference Relationships (HLPR) to improve the credibility of WSNs by fusing uncertain information and putting forward exact opinion about different WSN schemes. Wang et al. [19] proposed a novel data fusion algorithm inspired by hesitant fuzzy entropy to reduce the redundant sensory data transfer in WSN clustering. In this paper, we have utilized some ideas from state-of-the-art research and provided a detailed solution for optimally handling problems like energy consumption and network lifetime using a novel hesitant fuzzy entropy-based opportunistic clustering and data fusion scheme in WSN.

## 3. System Modeling

### 3.1. Network Model

In this paper, we considered a heterogeneous WSN in which all sensor nodes are deployed uniformly in a field bounded by $L\times L\;\left(m^{2}\right)$ region. Each static sensor node with a unique ID can acquire its neighbors’ ID by sharing a probe message as discussed in [22,26]. Also, we assumed that all sensor nodes follow the asynchronous working–sleeping cycle strategy with their working time as $W_{V}$ and sleeping time as $S_{V}$. In this model, we assumed that the BS has unlimited energy, storage space and powerful computation capability for communication between CHs and the BS to collect the aggregated sensory data and perform comprehensive evaluations for detection of an event. Furthermore, we also assumed that each sensor node has its own CH and each sensor node has limited energy and limited storage capacity.

Figure 1 illustrates our network model in which we have considered a single BS, some CHs and many sensor nodes connected to each CH. It is evident from Figure 1 that each sensor node acting as CMs or CHs and BS have fixed locations after deployment in the network. Each CH acquires the sensory data from its CMs. CHs can communicate with the BS or any other CH directly or through multiple hops. Moreover, every sensor node is aware of its position using the energy-efficient localization method and we can estimate the distance between the CM and CH and the CH and BS using the Relative Signal Strength Indicator (RSSI) parameter.

Anees et al. [26] utilized the OR in WSNs using the asynchronous working–sleeping cycle strategy for sensor nodes deployed in neighborhood area networks (NANs) of smart grids (SGs). For the efficient utilization of OR, we have to model it using the working–sleeping schedule ($W_{v}/S_{v}$), and the status transition frequency ($F_{ST}$) of each sensor node in the network. As the concept of OR is quite realizable in real-time scenarios of WSNs, we adopt the asynchronous working–sleeping cycle strategy for every sensor node deployed in our network. The list of notations is given in Table 1.

### 3.2. Energy Model

We considered three different types of sensor node in terms of energy levels—i.e., normal, intermediate and advanced. If $E_{n}$ is the initial energy of normal sensor nodes, then the initial energy of the intermediate and advanced sensor nodes should be higher than the normal sensor nodes by a factor of μ=(Eint−En)En and τ=(Eadv−En)En, respectively. Using this information, we can determine the total energy of the sensor nodes in Equation (1) as:(1)$$E_{total}=\;N\;(E_{n}+R_{int}(E_{int}-E_{n})\;+R_{adv}(E_{adv}-\;E_{n}))=\;NE_{n}\left(1+R_{int}\mu +\;R_{adv}\tau \right)$$
where N is the total number of sensor nodes in the network, $R_{int}$ is the ratio of intermediate sensor nodes and $R_{adv}$ is the ratio of advanced sensor nodes in the network. Furthermore, we considered the simplified energy consumption model based on [26] for radio energy dissipation during transmission and reception. According to this model, the energy required to transmit l bits of data over distance d can be given in Equation (2) as:(2)$$E_{T}\left(V_{i},V_{j}\right)=\;\{\begin{array}{c} E_{elec}l+\;\varepsilon_{fs}ld_{V_{i}V_{j}\;\;\;\;\;\;\;\;\;\;\;\;\;\;\;\;\;\;\;\;\;\;\;\;\;\;\;\;\;\;\;\;\;}^{2}d<d_{0}\\ E_{elec}l+\;\varepsilon_{mp}ld_{V_{i}V_{j}}^{4}\;\;\;\;\;\;\;\;\;\;\;\;\;\;\;\;\;\;\;\;\;d\geq d_{0} \end{array}$$
where $E_{elec}$ is the energy spent by transmitter on running the radio electronics, $\varepsilon_{fs}$ is the energy dissipated by power amplifier depending on the Euclidean distance $d_{V_{i}V_{j}}$ between the transmitter and receiver when free space fading is considered for distance less than $d_{o}$, and $\varepsilon_{mp}$ is the energy dissipated by power amplifier depending on Euclidean distance $d_{V_{i}V_{j}}$ between transmitter and receiver when multi-path fading is considered for distance greater than $d_{o}$. The threshold distance is given as do=εfs εmp. Similarly, the energy required to receive l bits of data over distance d can be computed in Equation (3) as:(3)$$E_{R}=\;E_{elec}l$$

Likewise, the energy spent by CM or CH at the beginning of each round for sensing l bits of data can be analyzed as $E_{sense}=\;E_{elec}l$. Here, we have assumed that the distance between CM and CH is shorter than the distance between CH and BS. Thus, we use $\varepsilon_{fs}$ (free space fading model) for communication between CM and CH and $\varepsilon_{mp}$ (multi-path fading model) for communication between CH and BS. Accordingly, the energy consumed by each CM can be calculated in Equation (4) as
(4)$$E_{CM}=\;E_{sense}+\;E_{T}=\;E_{elec}l+\;E_{elec}l+\;\varepsilon_{fs}ld_{V_{i}V_{j}}^{2}$$

Each CH aggregates the received sensory data and then forwards it towards BS. The energy consumed by each CH can be calculated in Equation (5) as
(5)$$\begin{array}{ll} E_{CH} & =E_{sense}+\;\left(\frac{N}{N_{C}}-1\right)E_{R}+\left(\frac{N}{N_{C}}\right)lE_{A}+\;\left(\frac{N}{r}\right)E_{T}\\ & =E_{elec}l+\;\left(\frac{N}{N_{C}}-1\right)E_{elec}l+\left(\frac{N}{N_{C}}\right)l\frac{E_{elec}}{R_{CC}}+\left(\frac{N}{r}\right)E_{elec}l\\ & +\;\left(\frac{N}{r}\right)\varepsilon_{mp}ld_{V_{i}V_{j}}^{4} \end{array}$$
where $N_{C}$ represents the number of clusters in the network, $\frac{N}{N_{C}}$ is the working sensor nodes per cluster in which we have 1 CH and $\frac{N}{N_{C}}-1$ CMs. $E_{A}$ represents the energy consumed during aggregating data packets at CH, r is the compression ratio and $R_{CC}$ is the communication to computation ratio used only during aggregation in this case. It is pertinent to mention here that the total energy consumed in a cluster is the sum of energies consumed by all CMs plus the energy consumed by their corresponding CH. The total energy consumed by a cluster $E_{cluster}$ is given in Equation (6) as
(6)$$E_{cluster}=\;E_{CH}+\;\overset{\left(\frac{N}{N_{C}}-1\right)}{\underset{i=1}{\sum }}E_{CM_{i}}$$

## 4. The Proposed Scheme HFECS

Information entropy is the statistical measure of the degree of uncertainty of an information source and it has always played a significant role in uncertainty decision analysis [13]. Likewise, hesitant fuzzy entropy is defined as the statistical measure of degree of hesitant fuzzy uncertainty, which depends on MADM to make a decision about certain events. In heterogeneous clustering schemes of WSNs, the role of CH within a cluster needs to be substituted using some decision criteria in order to avoid the hotspot problem. Using a single parameter such as the energy level of the sensor nodes is not sufficient enough to make an accurate decision about CH selection after every round of communication. In order to ensure that the best possible working sensor node is chosen as CH in every round of communication, we have to incorporate MADM in our clustering scheme. The hesitant fuzzy set including multi attributes and alternatives could help us out in implementing MADM for opportunistic clustering in WSNs [40,41].

Our main objective is to design a hesitant fuzzy entropy-based self-organizing opportunistic clustering scheme which can prolong the network lifetime and bring an improvement in overall energy consumption of sensor nodes. In our proposed scheme, we determined the entropy weight coefficients using hesitant fuzzy entropy to select the best possible CH through election procedure and perform reliable data fusion at CH level to reduce the information flow. Since we are dealing with MADM problem, we considered several parameters like time frequency parameter $TF_{V_{i}V_{j}}$ (depends on $W_{v}/S_{v}$ and status transition frequency $F_{ST}$), residual energy $R_{E}$, link quality factor *LQR* in terms of signal-to-noise ratio, distance to BS $D_{toBS}$, and number of supported sensor nodes $N_{support}$ as our different attributes of hesitant fuzzy set to compute the hesitant fuzzy entropy.

We have three different types of nodes in our network which can lead to energy heterogeneity and might result in three different types of CH, i.e., $CH_{normal}$,$CH_{int}$, and $CH_{adv}$, with the initial energy $E_{n}$, $E_{int}$, and $E_{adv}$, respectively, but in order to avoid frequent change in the CH role, we used only two different types of CH, i.e., $CH_{int}$ and $CH_{adv}$, for calculating $N_{support}$. However, the percentage of energy consumption for $CH_{int}$ ($E_{CH}/E_{int}$) is more than that of $CH_{adv}$ ($E_{CH}/E_{adv}$) due to the reason that $E_{adv}$ >$\;E_{int}$. As the consumed energy in relation to available energy is uneven for CHs, it leads us to determine the optimum number of sensor nodes which can be supported by each type of CH. Therefore, we used the remaining energy and average of $E_{int}$ and $E_{adv}$ to find the optimum number of sensor nodes supported by both $CH_{int}$ and $CH_{adv}$ in Equation (7) as:(7)$$N_{support}=\frac{\left(E_{total}-E_{consumed}\right)*\left(N-N_{C}\right)}{\left(N_{C}*\;\frac{\left(E_{int}+E_{adv}\right)}{2}\right)}$$

### 4.1. CH Election Procedure

Based on the sensory data being generated by sensor nodes, we make a decision about CH role in the cluster using $TF_{V_{i}V_{j}}$,$\;R_{E}$, *LQR*, $D_{toBS}$ and $\;N_{support}$. But before that, we have to construct the hesitant fuzzy entropy matrix and determine the corresponding weight coefficents of each attribute. Subsequently, the attribute values are synthesized to determine the threshold attribute values. The threshold attribute values are then compared with original attribute values of each sensor node. Based on their comparison, the BS selects the sensor nodes with original attribute values greater than threshold attribute values and add those sensor nodes in CH election set.

The entropy measures for hesitant fuzzy set were already discussed by Xu et al. in [40] and Xia et al. in [41]. Due to the fact that hesitancy is a common problem in decision making problems, we need to develop some entropy or cross entropy measures for hesitant fuzzy sets. Wang et al. in [19] discussed that if X is a fixed set and the hesitant fuzzy set α when applied on X returns a subset in [0,1]. Let $l_{\alpha \left(x\right)}$ be the number of values in α(x) where $\alpha {\left(x\right)}_{\sigma_{i}}$ be the ith smallest value in α(x) in which $i=\left\{1,2,\ldots l_{\alpha \left(x\right)}\right\}$. In order to find the cross entropy of two hesitant fuzzy sets, i.e., α and β, we assume that both of them have same length, or, if there is only one value in α, we extend it by repeating that value until it reaches the length of β. Moreover, if there is only one hesitant fuzzy set, then we use α and $\alpha^{C}$(complement of α) in the cross-entropy and calculate the hesitant fuzzy entropy in Equations (8) and (9) as
(8)$$E_{A}\left(\alpha \right)\;=\;1\;-\;C_{A}\left(\alpha ,\;\alpha^{C}\right)$$
where $C_{A}\left(\alpha ,\;\alpha^{C}\;\right)=\;\frac{2}{l_{\alpha }T}\overset{l_{\alpha }}{\underset{i=1}{\sum }}(\frac{\left(1+q\alpha_{\sigma \left(i\right)}\right)\ln\left(1+q\alpha_{\sigma \left(i\right)}\right)+\left(1+q\left(1-\alpha_{\sigma \left(l_{\alpha }-i+1\right)}\right)\right)\ln\left(1+q\left(1-\alpha_{\sigma \left(l_{\alpha }-i+1\right)}\right)\right)}{2}-$
$\frac{2+q\alpha_{\sigma \left(i\right)}\;q\left(1-\alpha_{\sigma \left(l_{\alpha }-i+1\right)}\right)}{2}\ln\frac{2+q\alpha_{\sigma \left(i\right)}\;q\left(1-\alpha_{\sigma \left(l_{\alpha }-i+1\right)}\right)}{2})\;q>0$
(9)$$\begin{array}{ll} E_{A}\left(\alpha \right) & =\;1\;-\;\frac{2}{l_{\alpha }T}\overset{l_{\alpha }}{\underset{i=1}{\sum }}(\frac{\left(1+q\alpha_{\sigma \left(i\right)}\right)\ln\left(1+q\alpha_{\sigma \left(i\right)}\right)+\left(1+q\left(1-\alpha_{\sigma \left(l_{\alpha }-i+1\right)}\right)\right)\ln\left(1+q\left(1-\alpha_{\sigma \left(l_{\alpha }-i+1\right)}\right)\right)}{2}\\ & -\frac{2+q\alpha_{\sigma \left(i\right)}\;q\left(1-\alpha_{\sigma \left(l_{\alpha }-i+1\right)}\right)}{2}\ln\frac{2+q\alpha_{\sigma \left(i\right)}\;q\left(1-\alpha_{\sigma \left(l_{\alpha }-i+1\right)}\right)}{2})\;q>0 \end{array}$$
where T=(1+q)ln(1+q)−(2+q)(ln(2+q)−ln(2)), q>0. Then, we calculate the entropy weight coefficients for all sensor nodes using Equation (10) after generating entropy matrix using Equation (9).
(10)$$w_{i}=\;\frac{1-E_{i}}{r-\sum_{i=1}^{r}E_{i}}$$
where $0\leq w_{i}\leq 1$, $\overset{r}{\underset{i=1}{\sum }}w_{i}=1$ and $E_{i}=\frac{1}{r}\;\overset{r}{\underset{i=1}{\sum }}E\left(\alpha_{ji}\right)$. In the next step, we find the average data value of attributes in measured value matrix, i.e., $D_{i}=\frac{1}{l_{\alpha }}\;\overset{l_{\alpha }}{\underset{i=1}{\sum }}\alpha_{\sigma \left(i\right)}$ and then synthesize the value of each attribute to find the threshold attribute value using Equation (11) as:(11)$$D_{Fi}=\frac{1}{r}\;\overset{r}{\underset{i=1}{\sum }}w_{i}D_{i}$$

The decision about placing sensor nodes in CH election set depends on the comparison between the original attribute values of sensor nodes in a cluster with that of threshold attribute values. A sensor node can be placed in the CH election set only if the maximum of its original attribute values is greater than the threshold attribute values generated from Equation (11). The pseudocode for this procedure is given in Algorithm 1.
**Algorithm 1.** CH Election Procedure in HFECS**Input:** Multi-Attribute values of each sensor node**Output:** Sensor nodes in CH election set**Begin:**1. r: number of attributes2. s: number of sensor nodes3. α: attribute value4. **for** i = 1 to r, **do**5.  **for** j = 1 to s, **do**6.   Measure the attribute values for all the sensors7.   Formation of measured value matrix8.   Standardize the attribute values of all sensors9.  **end for**10.  Generate the hesitant entropy decision matrix after standardization of each attribute value11.  Calculate the hesitant fuzzy entropy using $E_{A}\left(\alpha \right)\;=\;1\;-\;C_{A}\left(\alpha ,\;\alpha^{C}\right)$12.  Refer to Equations (8) and (9) for calculation of cross entropy $C_{A}\left(\alpha ,\;\alpha^{C}\right)$13.  Generate the entropy matrix based on calculated hesitant fuzzy entropy for all attributes14. **end for**15. Find the entropy value of each attribute using $E_{i}=\frac{1}{r}\;\overset{r}{\underset{i=1}{\sum }}E\left(\alpha_{ji}\right)$16. **for** i = 1 to r, **do**17.  Determine the weight value $w_{i}$ for all sensor nodes using Equation (10)18. **end for**19. Find the average data value in measured value matrix, i.e., $D_{i}=\frac{1}{l_{\alpha }}\;\overset{l_{\alpha }}{\underset{i=1}{\sum }}\alpha_{\sigma \left(i\right)}$20. **for** i = 1 to r, **do**21.  Synthesize the value of each attribute to find the threshold value using Equation (11)22.  **if** (sensor node with $\alpha_{\sigma \left(i\right)}$ > $D_{Fi}$), **then**23.   Add k sensor nodes with highest $(\alpha_{\sigma \left(i\right)}$ > $D_{Fi})$ in CH Election Set24.  **end if**25. **end for**

### 4.2. Cluster Formation

In HFECS, the BS executes the election procedure and creates a CH election set. Initially, when $C_{R}=1$, the BS adds k sensor nodes with highest values of $\alpha_{\sigma \left(i\right)}$ in the CH election set after executing the election procedure given in Algorithm 1 and then invokes those k sensor nodes by sending them a CH_Election_set message. After receiving the CH_Election_Set message from the BS, the potential CH starts preparing its CH_announce message and then send this message to all working neighbors in that cluster to advertise its role as a potential CH. The pseudocode for this procedure is given in Algorithm 2.
**Algorithm 2.** HFECS (BS Side)**Input:** CH election procedure**Output:** Potential CH advertising its role and making local decision for data fusion**Begin:**1. **if** ($C_{R}$ = 1), **then**2.  BS executes the CH election procedure3.  BS creates a CH election set4.  BS adds k sensor nodes in CH election set with highest $(\alpha_{\sigma \left(i\right)}$ > $D_{Fi})$5.  BS invokes k working sensors with highest attribute values in CH election set by sending6.  CH_Election_Set message7. **end if**8. Wait for collected data from CHs9. **if** (final prediction result received from CH), **then**10.  Evaluate the collected data to determine the possibility of event happening11.  **if** (event happened)12.   Perform standard operation procedure13.  **end if**14. **end if**

If the CH_announce message is sent by multiple potential CHs, then CMs send Connection Request ‘Con_Req’ message to the CH with highest attribute values $\alpha_{\sigma \left(4\right)}$ > $D_{F4}$. For example, if a neighboring sensor node receives CH_announce from multiple potential CHs, then the sensor node becomes a cluster member of that CH which offers better link quality with that neighboring sensor node in comparison to other CHs. However, if the CH_announce message is sent by a single potential CH, then CMs send a Con_Req message to that CH without any delay. Meanwhile, the new CH creates a CM_set and adds all the working CMs in CM_set after receiving the Con_Req messages from CMs. Subsequently, the new CH transmits the TDMA schedule to all CMs in that cluster, keeping in view the asynchronous working–sleeping cycle of CMs. This TDMA schedule for every CM should be selected vigilantly from a CH as the timeslot for forwarding the sensory data to the CH should be within working time of that CM. Each CM extracts its own CM timeslot from the TDMA schedule sent by the CH, stores the CM timeslot in its local buffer and waits until the CM timeslot becomes current timeslot. The pseudocode for CM task execution is given in Algorithm 3.

The construction of OCRG is very important to understand the asynchronous working–sleeping cycle of sensor nodes. Therefore, we need to construct an OCRG to analyze the opportunistic connections between CM-CH and CM-CM in a heterogeneous WSNs. When the CM timeslot becomes the current timeslot, $V_{CM\left(i\right)}$ calculates its link connectivity with $V_{CM\left(j\right)}$ in terms of time-frequency parameter, residual energy, LQR, distance from the BS and Nsupport as per Equation (12),
(12)$$L_{V_{CM\left(i\right)}V_{CM\left(j\right)}}=\begin{array}{c} max\\ V_{CM}\in S_{j} \end{array}\left\{\alpha TF_{V_{CM\left(i\right)}V_{CM\left(j\right)}}+\beta R_{E,V_{CM\left(i\right)}V_{CM\left(J\right)}}+\gamma S/N_{V_{CM\left(i\right)}V_{CM\left(j\right)}}+\;\sigma D_{toBS\left(CM\left(i\right)\right)}+\;\delta N_{support\left(CM\left(i\right)\right)}\right\}$$
where $i=j=1:\frac{N}{N_{C}}-2$, $TF_{V_{CM\left(i\right)}V_{CM\left(j\right)}}$ is the time-frequency parameter between $V_{CM\left(i\right)}$ and $V_{CM\left(j\right)}$, $R_{E,V_{CM\left(i\right)}V_{CM\left(j\right)}}$ is the residual energy of node $V_{CM\left(i\right)}$ and $V_{CM\left(j\right)}$, $S/N_{V_{CM\left(i\right)}V_{CM\left(j\right)}}$ is the link quality factor between $V_{CM\left(i\right)}$ and $V_{CM\left(j\right)}$, $D_{toBS\left(CM\left(i\right)\right)}$ is the distance to the BS from $V_{CM\left(i\right)}$ and $N_{support\left(CM\left(i\right)\right)}$ is the number of sensor nodes supported by $V_{CM\left(i\right)}$ in case $V_{CM\left(i\right)}$ becomes $V_{CH}$. α, β, γ, σ and δ are the appropriate weights assigned to time-frequency parameter, residual energy, link quality factor, distance to the BS and number of supported sensor nodes, respectively. The time-frequency parameter depends on working-sleeping cycle $W_{v}/S_{v}$ and status transition frequency $F_{ST}$. We can calculate the time-frequency parameter $TF_{V_{CM\left(i\right)}V_{CM\left(j\right)}}$ using Equation (13) as
(13)$$TF_{V_{CM\left(i\right)}V_{CM\left(j\right)}}=\;\left(\frac{F_{STV_{CM\left(i\right)}}}{F_{ST_{max}}}\;\times \frac{W_{V_{CM\left(i\right)}}}{T_{CP}}\right)\left(\frac{F_{STV_{CM\left(j\right)}}}{F_{ST_{max}}}\;\times \frac{W_{V_{CM\left(j\right)}}}{T_{CP}}\right)$$
where $W_{V_{CM\left(i\right)}}$ and $W_{V_{CM\left(j\right)}}$ are the total working time of the adjacent nodes $V_{CM\left(i\right)}$ and $V_{CM\left(j\right)}$, $T_{CP}$ is the data collection period, $F_{ST}{}_{CM\left(i\right)}$ and $F_{ST}{}_{CM\left(j\right)}$ are the status transition frequencies of the adjacent nodes $V_{CM\left(i\right)}$ and $V_{CM\left(j\right)}$, and $F_{ST}{}_{max}$ is the max status transition frequency value obtained during $T_{CP}$. Then, $V_{CM\left(i\right)}$ acquires the sensory data, integrates it while removing the redundant information, and forwards the integrated data along with the link connectivity information to the current CH.
**Algorithm 3.** HFECS (CM Side)**Input:** CH election procedure executed**Output:** Executing all tasks relevant to CM after CH election procedure**Begin:**1. **if** (CH_Announce message received from multiple potential CH), **then**2.  Send the Con_Req to potential CH with highest attribute value $\alpha_{\sigma \left(4\right)}$ > $D_{F4}$3. **else if** (CH_Announce message received from single potential CH), **then**4.  Send the Con_Req to that potential CH5. **else** Wait till the CH_Announce message is received from any potential CH6. **end if**7. Wait till the TDMA schedule is received from new CH8. **if** (TDMA based schedule received from new CH), **then**9.  Store the schedule (especially CM timeslot) in local buffer10.  **if** (timeslot = CM timeslot), **then**11.   Calculate Link connectivity of $V_{CM\left(i\right)}$ with $V_{CM\left(j\right)}$ using Equation (12)12.   Integrate data and remove the redundant and identical info. for every time period T13.   while $D_{T_{i}}\;\neq \;D_{T_{i-1}}$, **do** // detecting the change in integrated data14.    Transmit the integrated data $D_{T_{i}}$ to CH along with $L_{V_{CM\left(i\right)}V_{CM\left(j\right)}}$15.   **end while**16.  **else if** (timeslot ≠ CM timeslot), **then**17.   Wait until timeslot = CM timeslot18.  **end if**19. **end if**

Besides the benefits of hesitant fuzzy entropy in the CH election procedure of heterogeneous clustering scheme, it can also help in predicting an event in the heterogeneous clustering scheme and in reducing the number of transmissions in the network by applying reliable data fusion. Let us assume that CH requires time period T to transmit aggregated data packet to BS, and CM requires time period t to transmit the integrated data to CH where T=nt. Moreover, we considered context-based data forwarding, which means that when the CH receives the integrated data from $V_{CM\left(i\right)}$, it performs lossless aggregation by concatenating multiple data packets into single packet of specified length. In addition, CH also standardizes the received integrated data, calculates the hesitant fuzzy entropy and assign corresponding weight coefficients to integrated data of all sensor nodes. Then, CH synthesizes the attribute values of each sensor node to determine the threshold attribute values. After that, the threshold attribute values are compared with original attribute values to make a decision about data forwarding to the BS. If the original attribute values (current integrated data) are greater than threshold attribute values, it means that integrated data is changed after time period T, only then the CH transmits the result to the BS so that the BS can initiate the Standard Operating Procedure (SOP) accordingly against the event occurred in the cluster, but if the original attribute values are less than threshold attribute values, it means that the integrated data are not changed after time period T and CHs do not need to forward the integrated data to the BS. In this case, CHs should wait for time period t so that more sensory data are available.

After forwarding the aggregated data packets to BS, CH calculates the link connectivity with all of its CM as given in Equation (14),
(14)$$L_{V_{CH}V_{CM\left(j\right)}}=\begin{array}{c} max\\ V_{CM}\in S_{j} \end{array}\left\{\alpha TF_{V_{CH}V_{CM\left(j\right)}}+\;\beta R_{E,V_{CH}V_{CM\left(j\right)}}+\;\gamma S/N_{V_{CH}V_{CM\left(j\right)}}+\;\sigma D_{toBS\left(CH\right)}+\;\delta N_{support\left(CH\right)}\right\}$$
where $TF_{V_{CH}V_{CM\left(j\right)}}$ is the time-frequency parameter between $V_{CH}$ and $V_{CM\left(j\right)}$, $R_{E,V_{CH}V_{CM\left(j\right)}}$ is the residual energy of $V_{CH}$ and $V_{CM\left(j\right)}$, $S/N_{V_{CH}V_{CM\left(j\right)}}$ is the link quality factor between $V_{CH}$ and $V_{CM\left(j\right)}$, $D_{toBS\left(CH\right)}$ is the distance to the BS from $V_{CH}$ and $N_{support\left(CH\right)}$ is the number of sensor nodes supported by $V_{CH}$. In this paper, the decision about the next CH is based on the link connectivity information of current CH and CMs. For each $V_{CM\left(i\right)}$ in CM_set, we compare the link connectivity $L_{V_{CH}V_{CM\left(j\right)}}$ with that of $L_{V_{CM\left(i\right)}V_{CM\left(j\right)}}\left(i\neq j\right)$. If $L_{V_{CH}V_{CM\left(j\right)}}>\;L_{V_{CM\left(i\right)}V_{CM\left(j\right)}}$, we add $V_{CH}$ in CH_Election_Procedure set ζ but if $L_{V_{CH}V_{CM\left(j\right)}}<\;L_{V_{CM\left(i\right)}V_{CM\left(j\right)}}$, then we add $V_{CM\left(i\right)}$ to CH_Election_Procedure set ζ.

If the current $V_{CH}$ is the only sensor node in CH_Election_Procedure set ζ, then current $V_{CH}$ will be the new $V_{CH}$ for next round $C_{R+1}$ but if there is only one sensor node in the CH_Election_Procedure set ζ and $V_{CH}\;\notin \;\zeta$, then the sensor node $V_{CM}$ present in ζ will be the new $V_{CH}$ for the next round $C_{R+1}$. However, if sensor nodes present in ζ are more than 1, then the current CH executes the CH election procedure using hesitant fuzzy entropy and selects the $V_{CM}$ based on highest $\left(\alpha_{\sigma \left(i\right)}>D_{Fi}\right)$ criteria. Subsequently, the current CH sends the CH_Election_Set message to the new CH. Upon receiving the CH_Election_Set message, the new CH needs to multicast the CH_announce message to its CMs in order to advertise its role as a potential CH. The pseudocode for CH task execution is given in Algorithm 4.
**Algorithm 4.** HFECS (CH Side)**Input:** CH election procedure executed by BS**Output:** CH decision and local data fusion decision using HFECS**Begin:**1. **if** (CH_Election_Set message received), **then**2.  Start preparing CH_Announce message in order to advertise your role as potential CH3.  Send CH_Announce message to neighboring sensor nodes $S_{j}$ which are in working mode4. **end if**5. Wait till the Con_Req message is received from neighboring $S_{j}$6. **if** (Con_Req message is received from neighboring $S_{j}$), **then**7.  CH creates a cluster member set, i.e., CM_set8.  Add the neighboring $S_{j}$ in CM_set9. **end if**10. **if** (all working $S_{j}$ in a cluster are part of CM_set of their new CH), **then**11.  Each new CH transmits the TDMA schedule to all CMs keeping in view the12.  working-sleeping cycle of CMs13. **end if**14. **if** (integrated data received from $S_{j}$|$S_{j}$
∈ CM_set), **then**15.  CH performs aggregation on multiple data packets received from $S_{j}$ by concatenating16.  into a single file of specified length17.  CH standardizes the data received from $S_{j}$, calculates the hesitant fuzzy entropy and weights18.  CH synthesize the attribute values of sensor nodes to determine threshold attribute values19.  CH compares the original attribute value with that of threshold attribute values20.   **if** (original attribute values are greater than threshold attribute values after T), **then**21.    CH periodically transmits the aggregated data to BS22.   **else if** (original attribute values are less than threshold attribute values after T), **then**23.    CH will not transmit the aggregated data to BS24.   **end if**25. **end if**26. Calculate Link connectivity of CH with its cluster members using Equation (14)27. for each cluster member $V_{CM\left(i\right)}$ in CM_set, **do**28.  **if** ($L_{V_{CH}V_{CM\left(j\right)}}$ ≥ $L_{V_{CM\left(i\right)}V_{CM\left(j\right)}}$), **then**29.   Add $V_{CH}$ to CH_Election_Procedure set ζ30.  **else if** ($L_{V_{CH}V_{CM\left(j\right)}}$ < $L_{V_{CM\left(i\right)}V_{CM\left(j\right)}}$), **then**31.   Add $V_{CM\left(i\right)}$ to CH_Election_Procedure set ζ32.  **end if**33. **end for**34. **if** (ζ = 1) and $V_{CH}\in \;\zeta$, **then**35.  $V_{CH}$ will be new $V_{CH}$ for next round $C_{R+1}$36. **else if** (ζ = 1) and $V_{CH}\notin \;\zeta$37.  $V_{CM}$ will be new $V_{CH}$ for next round $C_{R+1}$38. **else if** (ζ > 1)39.  $V_{CH}$ executes the CH election procedure and selects new CH with highest $\left(\alpha_{\sigma \left(i\right)}>\;D_{F_{i}}\right)$40.  $V_{CH}$ sends CH_Election_Set message to new CH41. **end if**

### 4.3. Packet Overhead Calculation

In our proposed scheme, each CH acquires the integrated sensed data from its CMs and then forwards the aggregated data to BS after detecting the change in integrated data for every time period *T*. In our paper, the hesitant fuzzy entropy balances the network load during CH rotation in successive rounds of communication when applied on MADM. Let us assume that we have $O\left(N_{W}\right)$ exchanged packets as an overhead in which $N_{W}$ is the number of working sensor nodes in the network. If we assume that k is the number of CHs per round, then we can calculate the overhead packets as:
k packets used for broadcasting CH_Announce by all CHs.$N_{W}-k$ packets in terms of Con_Req by working CMs.k packets used for broadcasting TDMA schedule to working CMs in CM_set by all CHs.$N_{W}-k$ packets used for sending data to CHs by working CMs.k packets used for forwarding data to the BS by all CHs in case if final prediction result is changed after time period *T*.$\frac{N}{N_{C}}-1$ packets received by all CHs from their CMs for link connectivity calculation.$\frac{N}{N_{C}}-1$ packets received by all CMs from their neighboring CMs and current CH for link connectivity calculation comparison.*k* packets used for the CHs rotation in case if all CHs select one of its CMs as the next CH.

Therefore, the overhead of HFECS will be $O\left(N_{W}\right)$ as $k\ll \frac{N}{N_{C}}\ll \;N_{W}$.

## 5. Case Study: HFECS in Neighborhood Area Networks

In this paper, we have described the opportunistic clustering mechanism based on hesitant fuzzy entropy using several parameters, such as the time-frequency parameter, residual energy, LQR, distance from the BS and Nsupport. Anees et al. in [26] proposed an energy-efficient multi-disjoint path opportunistic node connection routing protocol for smart grids in which the OCRG was utilized to calculate the link and path connectivity between sensor nodes for data forwarding purposes. Likewise, we have considered the neighborhood area networks (NANs) of smart grids in our case study to prove the significance of hesitant fuzzy entropy in heterogeneous clustering of WSNs. If we make an analogy between heterogeneous clustering and NANs of SGs, we can consider the NAN gateway as the CH and the several Home Area Network (HAN) gateways connected to the NAN gateway as CMs connected to their CH, as shown in Figure 2. Furthermore, the Access gateway in NANs can be considered as the BS, which receives the aggregated data from multiple NAN gateways and forwards it to the control center [46]. In this case study, we can see how the efficient use of hesitant fuzzy entropy can help us in making the CH election decision and also perform reliable data fusion.

We have selected the measured values of six attributes from five sensor nodes placed randomly in a cluster. It is important to mention here that for this case study we considered the network area as centralized NAN for SGs communication. As each of the attributes have different [min max] values, so we needed to perform the data standardization in order to bring all the attributes on the same scale and to convert our measured data values into hesitant fuzzy set. So, we divided the attribute values into 10 parts according to their [min max] values and then assign values corresponding to 0–1.0. Table 2 represents the interval for each attribute used for data standardization whereas Table 3 represents the measured value of attributes for all sensor nodes. Different monitoring values for each attribute indicate that data acquisition cycle for each sensor node is different due to the asynchronous working-sleeping cycle strategy. For example, for the attribute time-frequency parameter in Table 3, its measured values are normalized to
{([0,10] → [0,0.1]), ([10,20] → [0.1,0.2]), …([60,70] → [0.6,0.7]), …([90,100] → [0.9,1])}. Similarly, for residual energy, the standardized attribute values will be {([0,0.25] → [0,0.1]), ([0.25,0.5] → [0.1,0.2]), …([1.5,1.75] → [0.6,0.7]), …([2.25,2.5] → [0.9,1])}. The procedure of data standardization is the same for all other attributes mentioned in Table 3, along with their intervals, but for attribute ‘Distance to BS’, the standardized values are {([150,136] → [0,0.1]), ([136,122] → [0.1,0.2]), ……([66,52] → [0.6,0.5])…([24,10] → [0.9,1.0])} because if ‘Distance to BS’ is minimum, then we can have better connectivity with BS.

Table 4 shows the corresponding hesitant fuzzy sets after the data standardization process. We utilized the hesitant fuzzy sets to compute hesitant fuzzy entropy using Equation (12) to acquire the statistical measure of uncertainty in our decision analysis for the CH election procedure and data fusion by CH. Table 5 represents the hesitant fuzzy entropy of each attribute used in our case study. After generating the hesitant fuzzy entropy matrix, we determined the entropy weight coefficients using Equation (13). Table 6 represents the entropy weight coefficients of all sensor nodes, i.e., $W_{i}={\left[W_{1,}W_{2},\ldots W_{r-1},W_{r}\right]}^{T}$

Moreover, the average values of all attributes in Table 2 and entropy weight coefficients in Table 6 were used to compute the threshold attribute values using Equation (14). As we have six attributes in total, so the six threshold attribute values in our case study are $D_{F_{1}}=59.6252,\;D_{F_{2}}=1.8748,\;D_{F_{3}}=81.8583,\;D_{F_{4}}=12.5952,\;D_{F_{5}}=54.7839,\;D_{F_{6}\;}=28.9909$. Consequently, we compare the measured attribute value of each sensor with that of theshold attribute value in Table 7. Here, we considered the attributes with the three highest entropy coefficient weights, i.e., $TF_{V_{i}V_{j}}$, $D_{toBS}$ and $\;N_{support}$.

According to Table 7, only three sensors fulfill our criteria after comparing the measured attribute values with threshold attribute values. The BS puts the sensor nodes with highest $(\alpha_{\sigma \left(i\right)}$ > $D_{Fi})$ in CH election set and then invokes those sensor nodes in the CH election set. Hence, in this case study, only S5 will be selected as the CH, but in reality, the whole process is much more complex than the one mentioned in this case study due to interference, overhead and energy constraints.

## 6. Performance Evaluation

### 6.1. Simulation Environment

In order to validate the effectiveness of HFECS, we evaluate the performance of our proposed scheme in MATLAB R2018a simulator using cross platform libraries such as MEX-API for simulating WSNs [47]. This Application Programming Interface (API) can provide the user with an easy bidirectional connection interface between MATLAB and OMNET. In our simulation environment, 150 sensor nodes were randomly placed in a variable network area, i.e., 100 × 100, 150 × 150, 200 × 200, 250 × 250, 300 × 300 and 350 × 350 m^2^ on a 2-D plane with the BS as the network center. Here, we assumed that the BS has no energy constraints due to unlimited power supply but the randomly deployed sensors have limited battery power and we need asynchronous working-sleeping scheduling for their energy replenishment. The parameters used in this simulation are defined in Table 8.

In our simulation, CHs send aggregated data to the BS after every T = 100 s and CMs send integrated sensed data to CH after every t = 20 s where n=Tt=5. We calculated the total energy of our network in terms of $R_{normal}$,$R_{int}$, $R_{adv}$ and $E_{n},\;E_{int}\;\mathrm{and}\;E_{adv}$ as 288.11 J. We utilized the energy model defined for our MAC layer (IEEE 802.15.4) to compute the energy consumed during data transmission, reception and sensing. Additionally, we also utilized the IEEE 802.15.4 MAC Layer specifications for data rate and data packet size. It is pertinent to mention here that we considered various studies for acquiring the parameters like $E_{A}$,$\;\varepsilon_{fs},\varepsilon_{mp}$ and energy consumed by CHs during data fusion [19,39]. The performance of our proposed scheme is evaluated by comparing it with various routing protocols like SEP-E [34], DEEC-E [37], EBCS [39] and TEEN [38] based on the performance metrics like stability period, half-life time, average residual energy, network lifetime and PDR.
Stability period—can be defined as the time difference between two specific points. The first point is when the simulation starts and the second point is when first sensor node dies.Half-life time—can be defined as the time duration from start of the network simulation until the time when 50% of the sensor nodes have no residual energy left to continue their data delivery tasks.Energy Consumption—can be defined as the total energy consumed in the network during data transmission, reception, sensing, status transitions (CM→CH) and aggregation functions.Network Lifetime—this parameter is used to demonstrate the complete lifetime of sensor nodes per communication round for varying network size in terms of stacked group bar chart.Packet Delivery Ratio (PDR)—defined as the ratio of total number of successfully received data packets at BS corresponding to total number of data packets generated in the network by all sensor nodes.

### 6.2. Simulation Results

The performance metrics like the stability period, half-life time, average residual energy, network lifetime and PDR are analyzed against two parametric benchmarks viz. network size and number of communication rounds. We considered the number of communication rounds up to 4000 in our simulation.

#### 6.2.1. Stability Period

Figure 3 depicts the stability period for SEP-E, DEEC-E, EBCS, TEEN and HFECS against different network sizes. It can be observed from Figure 3 that the stability periods of HFECS and EBCS outperform all other clustering schemes as both of them are entropy-based clustering schemes. Furthermore, the stability period of all clustering schemes is decreased with the increase in network size due to the reason that the distance between sensor nodes and the energy consumption is increased as we increase our network size. For most of the clustering schemes, energy is consumed for performing data forwarding and lossy aggregation tasks but in the case of HFECS, we perform lossless aggregation at CHs by concatenating multiple data packets into a single large data packet of specified length to reduce the energy consumption, thus resulting in an enhanced stability period in comparison to other clustering schemes for larger network sizes. As DEEC-E works on the basis of absolute threshold value calculated in each round, the residual energy of all sensor nodes is then compared with this threshold value. As this threshold value does not take into account the network size, the stability period of this scheme is significantly reduced when the network size increases up to 350 × 350 m^2^. The degraded performance of SEP-E in comparison to EBCS and HFECS can be explained as the probability of each type of sensor node do not consider the remaining energy while selecting the new CH. Also, when the network size increases, the low density regions in the network have a greater number of CHs, which results in the inefficient utilization of energy by the CMs and correspondingly the stability period of SEP-E is reduced. The performance of EBCS in terms of the stability period is better than SEP-E, DEEC-E, and TEEN for all the network sizes.

The performance of EBCS for network sizes up to 200 × 200 m^2^ is slightly better than for HFECS. The reason for this is that when sensor nodes supporting the asynchronous working–sleeping cycle are deployed in a smaller network area, we have a high density of sensor nodes in every cluster, i.e., greater number of cluster members to be facilitated by their assigned CH, so it is highly likely that a potential CH or a CM will lose all of its residual energy in a slightly less time than EBCS due to the extra computation of link connectivity calculations performed in every cluster. This extra computation is due to the increased number of sensor nodes present in every cluster. As we increase the network size, the uniformly random deployment of sensor nodes create clusters with a lower number of sensor nodes as compared to the scenario of smaller network sizes, so the probability of a sensor node in HFECS for losing all of its energy decreases in comparison to the probability of a sensor node operating in EBCS. Table 9 shows the percentage improvement in terms of the stability period.

#### 6.2.2. Half-Life Time

Figure 4 depicts the half-life time of SEP-E, DEEC-E, EBCS, TEEN, and HFECS against different network sizes. It can be seen from Figure 4 that HFECS and EBCS enhance the sensor nodes’ half-life time in comparison to SEP-E, DEEC-E, and TEEN. Since the entropy weight coefficients are updated in every communication round, we see an improvement in half-life time and correspondingly better network lifetime. Moreover, as the number of sensor nodes which can be supported ($N_{support}$) by a CH implicitly reflects the cluster load which the CH can handle, so we included $N_{support}$ as one of our significant attributes in CH decision making process.

The performance of EBCS in terms of half-life time is better than SEP-E, DEEC-E, and TEEN for all the network sizes. In addition, EBCS achieves better half-life time as compared to HFECS for network sizes up to 200 × 200 m^2^ due to an increase in the cluster load and the extra computation required for calculating link connectivity between sensor nodes in the case of HFECS. For larger network sizes, HFECS performs better than EBCS due to the decrease in the cluster load, involvement of multiple attributes in decision making processes and sensor nodes with opportunistic connections, which finally results in a slight performance inversion between HFECS and EBCS. The percentage improvement of HFECS in terms of half-life time against different clustering schemes is given in Table 10.

#### 6.2.3. Average Residual Energy

We computed the average residual energy in two ways, i.e., (i) against different network sizes while keeping the communication round as constant, (ii) against the number of communication rounds while keeping the network size as constant. Also, we have computed the average residual energy against different network sizes at 1000 rounds and 2000 rounds to check the performance of all clustering schemes. Figure 5 depicts the average residual energy of SEP-E, DEEC-E, EBCS, TEEN, and HFECS for different network sizes at 1000 rounds. It can be observed from Figure 5 that the performance behavior of all clustering schemes is analogous at different network sizes. Furthermore, HFECS outperforms all other clustering schemes including EBCS for larger network sizes. As we increase the network size, the average residual energy starts dropping down for all clustering schemes, which indicates the increase in energy consumption. A noticeable increase in energy consumption can be seen from SEP-E, EBCS, and TEEN but there is a remarkable increase in energy consumption for DEEC-E, due to higher negative slope in average residual energy. Likewise, Figure 6 depicts the average residual energy of SEP-E, DEEC-E, EBCS, TEEN and HFECS for different network sizes at 2000 rounds.

Figure 7 illustrates the average residual energy of SEP-E, DEEC-E, EBCS, TEEN and HFECS against number of communication rounds while keeping the network size as 300 × 300 m^2^. The performance behavior of all clustering schemes is analogous at different numbers of communication rounds. However, DEEC-E and HFECS perform slightly better than other clustering schemes after the 1500th round of communication. DEEC-E functionality depends on an absolute threshold value calculated in every round which do not involve network size, so the average residual energy of DEEC-E is reduced as we increase the communication rounds due to sensor nodes’ poor lifetime and the increase in network size, but comparing the remaining energy of sensor nodes with that of an absolute threshold value gives advantage to sensor nodes with energy slightly above the threshold value, which results in better average residual energy than the rest of the clustering schemes. For smaller network sizes, the residual energy of the sensor nodes in HFECS is also consumed by link connectivity calculations for a higher number of sensor nodes per cluster, therefore the average residual energy of DEEC-E for smaller network sizes is better than in HFECS, but, as we increase the network size, the cluster load in HFECS reduces, thus resulting in better average residual energy for HFECS. Furthermore, as HFECS performs lossless data aggregation at the CH node for reducing the information flow and energy consumption, we have better average residual energy for higher values of communication rounds for HFECS than other clustering schemes. The percentage improvement of HFECS in terms of average residual energy is given in Table 11.

#### 6.2.4. Network Lifetime

Network lifetime can be defined as the time interval between the start of network simulation and when the first sensor node dies, all sensor nodes die, some percentage of sensor nodes die, loss of coverage occurs or network is partitioned such that path from source to sink does not exist [26]. According to [26], network lifetime can also be evaluated in terms of the dead node ratio in the network. The time when 25% or 50% of the nodes present in the network have no residual energy left to continue their data delivery tasks can be treated as the network lifetime. Here, we considered two such time intervals, i.e., (i) when the first sensor node dies (FND) and (ii) when half the nodes in the network die, i.e., 50% of the sensor nodes have no residual energy left to continue their data sensing tasks (HND). The stacked grouped bar chart in Figure 8 demonstrates the network lifetime in terms of two different cases, i.e., First Node Dead (FND) and Half Nodes Dead (HND) for SEP-E, DEEC-E, EBCS, TEEN and HFECS against different network sizes. In every stack, we have five bars representing five different schemes, i.e., SEP-E, DEEC-E, EBCS, TEEN and HFECS. In each bar, we have two groups, i.e., FND and HND.

It can be observed from Figure 8 that the performance behavior of all clustering schemes is analogous for network lifetime, i.e., a decreasing trend as we increase the network size. For smaller network sizes, the network lifetime is higher for all clustering schemes as the nodes located in close vicinity are easy to communicate with but, for larger network sizes, the network lifetime decreases rapidly due to the fact that sensor nodes are not located in the close vicinity and extra energy is being consumed for communication between sensor nodes, which results in decreased network lifetime. The performance of EBCS for network sizes of up to 200 × 200 m^2^ is slightly better than HFECS. For smaller network sizes, we have a high density of sensor nodes in every cluster, so it is highly likely that a potential CH or a CM lose all of their residual energy in slightly less time than EBCS due to the extra computation of link connectivity calculations performed in every cluster. However, as we increase the network size, the uniformly random deployment of sensor nodes creates clusters with a lower number of sensor nodes as compared to the scenario of smaller network sizes, thus resulting in better network lifetime in comparison to EBCS. HFECS also adopts lossless data aggregation in multiple data packets at the CH level, which reduces the information flow from CHs to the BS, thus decreasing some of the extra energy spent on control packet overhead. The percentage improvement of HFECS in terms of network lifetime (FND and HND) is given in Table 12 and Table 13.

#### 6.2.5. Packet Delivery Ratio (PDR)

The higher percentage of successfully received packets at the BS reflects the higher network reliability. In the case of HFECS, multiple data packets received by every CH node are aggregated into a single packet of specified length and then sent to the BS, so we have to incorporate aggregated data packets in PDR calculation instead of normal data packets. The percentage of successfully received aggregated data packets at the BS corresponding to total number of packets generated by all sensor nodes will be the PDR, in the case of HFECS, i.e.,
(15)PDR (Aggregated)=(∑i=1nADPiTotal Data packets generated )∗100
where *n* is the total number of aggregated data packets received at the BS and $ADP_{i}$ denotes the aggregated data packet received by BS at the $i^{th}$ instance in Equation (15). Figure 9 shows the PDR for SEP-E, DEEC-E, EBCS, TEEN and HFECS corresponding to different rounds of communication with our network size as 300 × 300 m^2^. As in the SEP-E protocol, the optimum number of CHs is not guaranteed due to its poor stability and every CH in SEP-E depends on the probability of each type of sensor node without considering the residual energy of sensor nodes, so it results in the reduction in average residual energy and network lifetime, thus leading to degraded PDR in comparison to HFECS, DEEC-E, and EBCS. The TEEN protocol performs worst against SEP-E, DEEC-E, EBCS and HFECS due to the low average residual energy and lower network lifetime with increasing communication rounds. For lesser communication rounds, HFECS shows degraded performance in comparison to EBCS due to the extra energy spent on computation purposes which result in status transitions of a few sensor nodes, thus reducing the PDR. However, for higher communication rounds, our proposed protocol, HFECS, outperforms all other clustering schemes due to its higher network lifetime, higher average residual energy for larger network sizes, data aggregation performed at the CH, and collision avoidance achieved through the assignment of TDMA slots to every CM, thus resulting in the increase in success rate of aggregated data packets at BS. The percentage improvement of HFECS in terms of PDR can be seen in Table 14.

## 7. Discussion

We have introduced the hesitant fuzzy entropy-based opportunistic clustering and data fusion scheme for WSNs. The previous entropy-based clustering schemes were designed without consideration of multiple attributes (especially asynchronous working-sleeping cycle) involved in the data flow from sensor nodes to CHs and to the BS. HFECS utilizes multiple attributes like the time frequency parameter, residual energy, remaining buffer capacity, link quality factor, the distance to the BS and the number of supported sensor nodes etc. in terms of local information of sensor nodes to measure hesitant fuzzy entropy and its corresponding weight coefficients for the CH election procedure and reliable data fusion, which improves the energy utilization and also increases the network lifetime of sensor nodes. The proposed designed of HFECS could be applicable to the following different approaches in future:
Designing a HFE-based multipath opportunistic routing protocol for wireless cognitive sensor networks.Designing a HFE-based clustering scheme with ring routing protocol.Unmanned Aerial Vehicle (UAV) sensor networksInternet-of-Things (IOT)-enabled home energy systemsCyber-physical systemsEnergy harvesting wireless sensor networks [48].Hesitant fuzzy entropy analysis in a density-based and grid-based opportunistic clustering scheme.

## 8. Conclusions

In this paper, a novel hesitant fuzzy entropy-based opportunistic clustering and data fusion scheme is proposed, utilizing the OCRG theory. Our proposed scheme, HFECS, achieves two levels of hierarchy in the network and energy heterogeneity is characterized using three levels of energy in sensor nodes. Self-organizing HFECS uses multiple attributes in terms of the local information of sensor nodes to measure hesitant fuzzy entropy and its corresponding weight coefficients for CH election procedure and reliable data fusion at CH to reduce the redundant information flow from CH to BS, which in turn improves the energy utilization and increases the network lifetime of sensor nodes. The performance of HFECS is evaluated against existing energy-efficient and entropy-based heterogeneous clustering schemes for parameters such as half-life period, stability period, average residual energy, network lifetime (FND and HND), and PDR. Our simulation results clearly show that HFECS performs better than the existing benchmarks for larger network sizes. Possible future work could include HFE-based ring routing in WSNs, HFE-based heterogeneous clustering supporting sink mobility, HFE based on Cognitive Radio Sensor Networks (CRSN) etc. In addition to this, the proposed scheme can be applied to ring routing [49], big data systems [50], data compression [51] and blockchain technology [52] in terms of reliable data fusion.

## Figures and Tables

**Figure 1 sensors-20-00913-f001:**
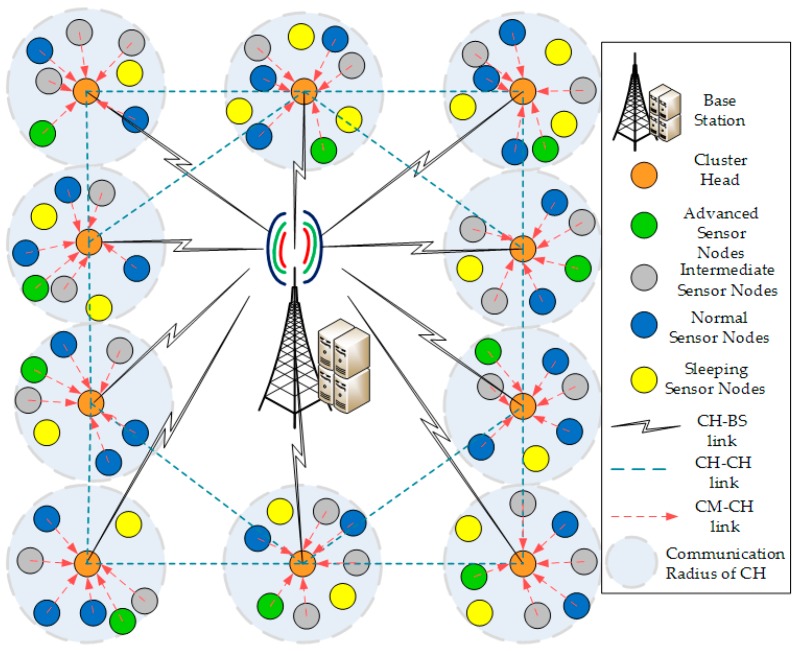
Network Model: heterogeneous wireless sensor networks (WSN) with three types of sensor nodes.

**Figure 2 sensors-20-00913-f002:**
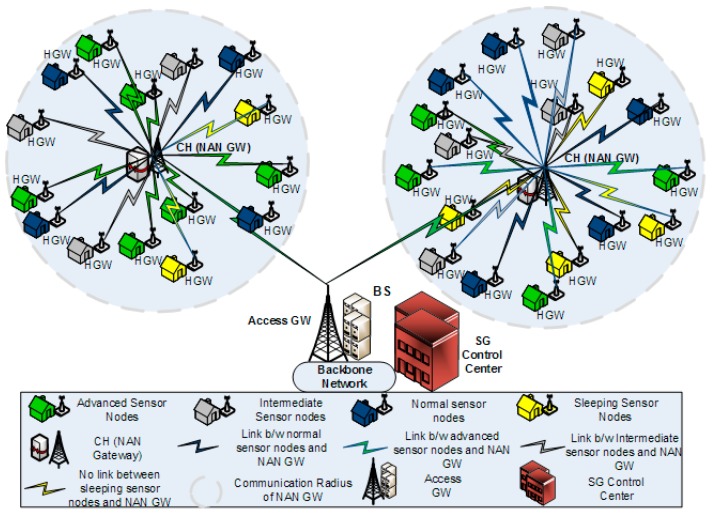
Hesitant fuzzy entropy (HFE) based opportunistic clustering for neighborhood area networks.

**Figure 3 sensors-20-00913-f003:**
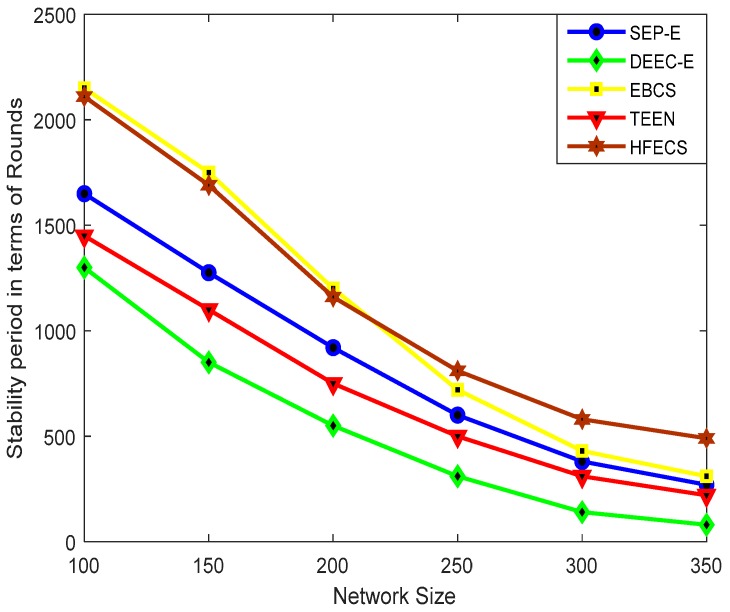
Stability period in terms of rounds against variable network size.

**Figure 4 sensors-20-00913-f004:**
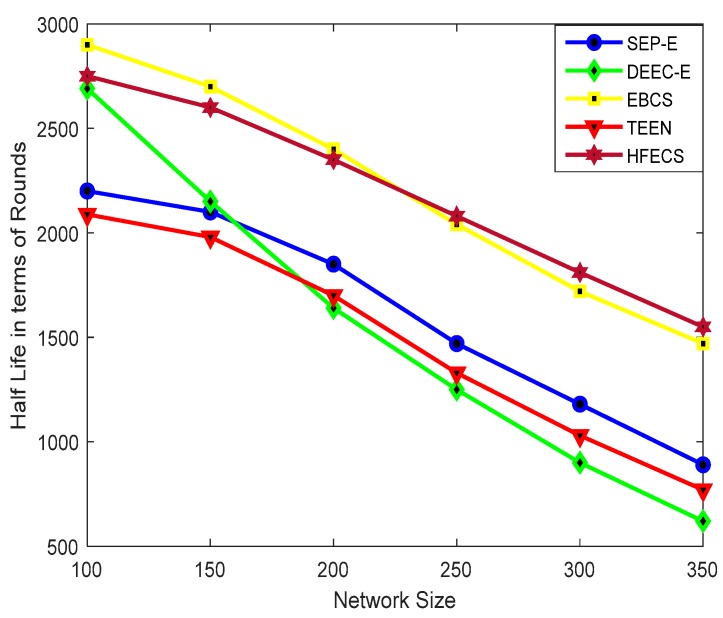
Half-life in terms of rounds against variable network size.

**Figure 5 sensors-20-00913-f005:**
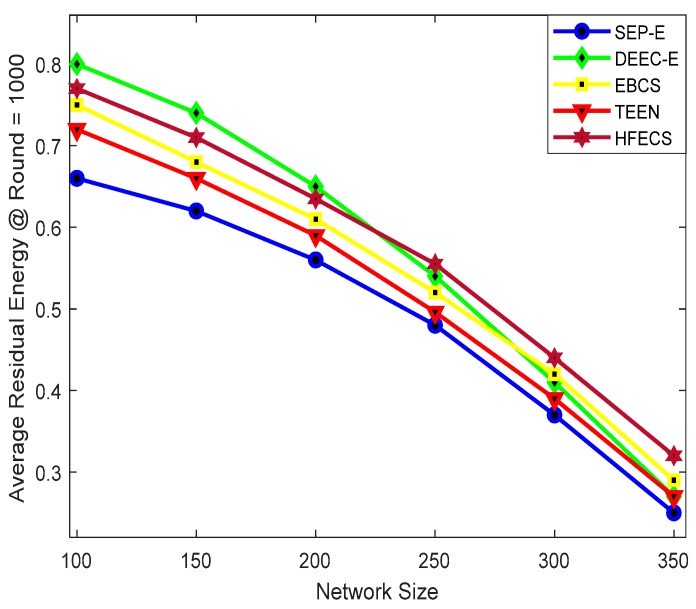
Average residual energy at round=1000 against variable network size.

**Figure 6 sensors-20-00913-f006:**
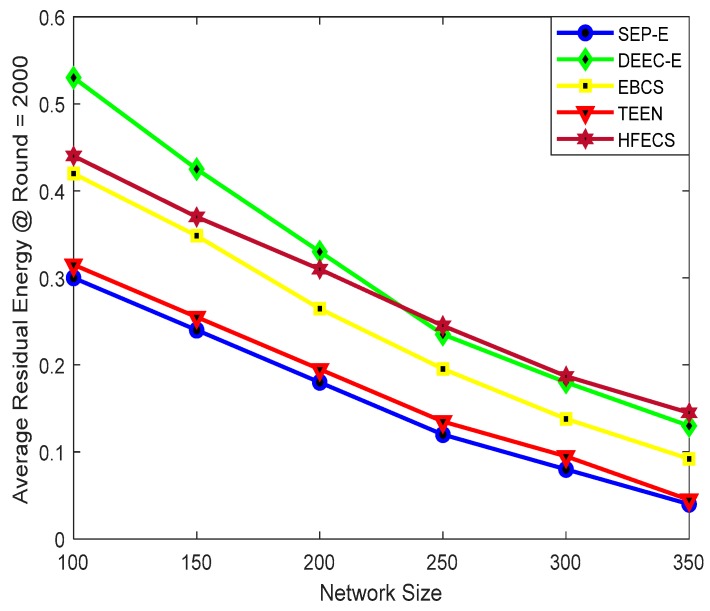
Average residual energy at round=2000 against variable network size.

**Figure 7 sensors-20-00913-f007:**
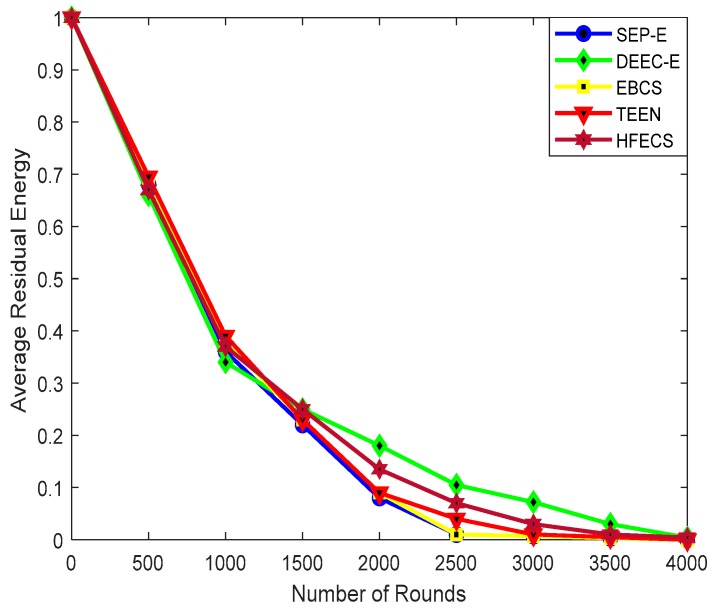
Residual energy against number of communication rounds.

**Figure 8 sensors-20-00913-f008:**
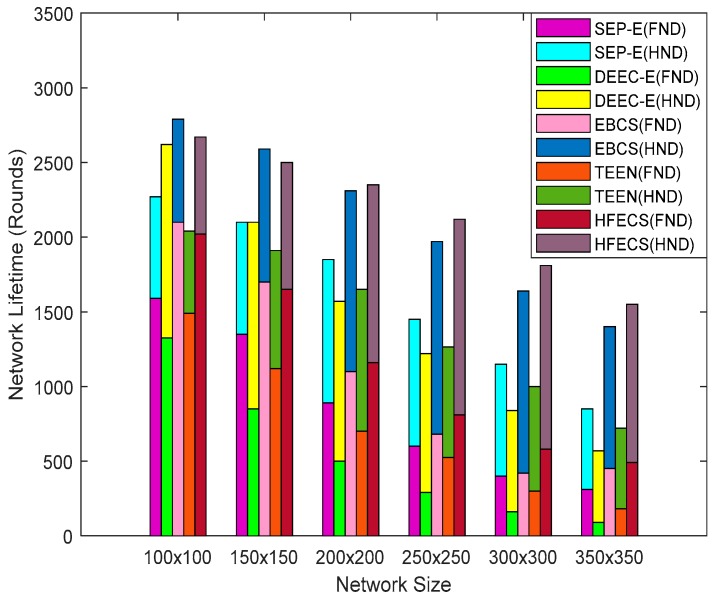
Network lifetime in terms of first node dead (FND) and half nodes dead (HND) against variable network size.

**Figure 9 sensors-20-00913-f009:**
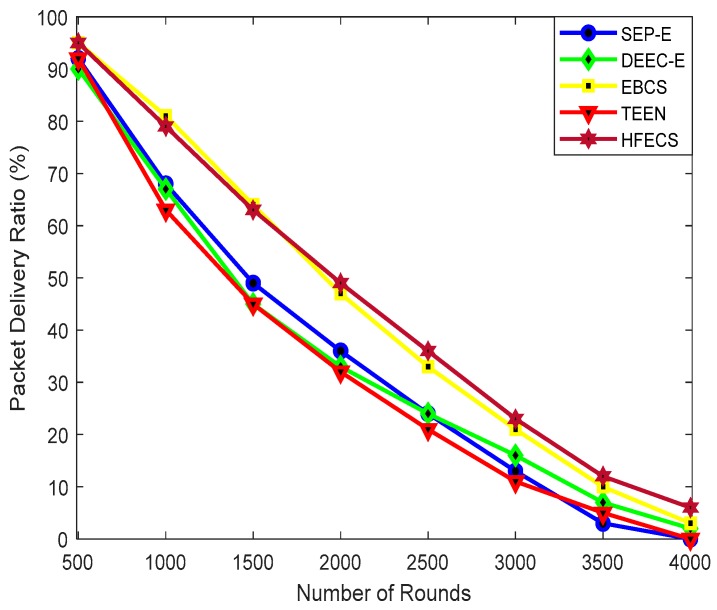
Packet delivery ratio (PDR) (%) against number of communication rounds.

**Table 1 sensors-20-00913-t001:** Notation description.

Notation	Meaning	Notation	Meaning
$E_{n}$	Initial energy of normal sensor nodes	$\;R_{L}$	Long-radio range
$E_{int}$	Initial energy of intermediate sensor nodes	$W_{v}/S_{v}$	Working/Sleeping schedule
$E_{adv}$	Initial energy of advanced sensor nodes	$F_{ST}$	Status transition frequency
μ	Energy weight factor of intermediate sensor nodes	$TF_{V_{i}V_{j}}$	Time-Frequency parameter
τ	Energy weight factor of advanced sensor nodes	$\;R_{E}$	Residual Energy of a sensor
$R_{int}$	Ratio of intermediate sensor nodes	*LQR*	Link quality factor(S/N)
$R_{adv}$	Ratio of advanced sensor nodes	$D_{toBS}$	Distance to Base Station
$E_{elec}$	Energy spent for running radio electronics	$\;N_{support}$	Number of supported sensors
l	Number of bits	$E_{A}\left(\alpha \right)$	Hesitant Fuzzy Entropy
$d_{V_{i}V_{j}}$	Euclidean distance b/w transmitter and receiver	$C_{A}\left(\alpha ,\;\alpha^{C}\right)$	Cross Entropy
$\varepsilon_{fs}$	Energy dissipation in free space fading	$w_{i}$	Entropy coefficient weight
$\varepsilon_{mp}$	Energy dissipation in multi-path fading	$D_{Fi}$	Final data attribute value
N	Number of sensor nodes	$\;C_{R}$	Communication round
$N_{C}$	Number of clusters	$\alpha {\left(x\right)}_{\sigma_{i}}$	ith value in fuzzy set
$E_{sense}$	Energy consumed during sensing	$l_{\alpha \left(x\right)}$	Length of fuzzy set
$E_{A}$	Energy consumed during aggregation	*α, β, γ, σ δ*	Weights for $TF_{V_{i}V_{j}}$, $R_{E}$, *LQR,* $D_{toBS}$ and $\;N_{support}$
r	Compression ratio	$L_{V_{CM\left(i\right)}V_{CM\left(j\right)}}$	Link connectivity b/w CM(i) and CM(j)
$R_{CC}$	Communication–Computation ratio	$L_{V_{CH}V_{CM\left(j\right)}}$	Link connectivity b/w CH and CM(j)
$E_{CH}$	Energy consumed by cluster head	$T_{CP}$	Data Collection period
$E_{CM}$	Energy consumed by cluster member	ζ	CH Election procedure set
$E_{cluster}$	Energy consumption of a cluster	T	Time period for sending aggregated data to BS by CH
$R_{S}$	Short-radio range	t	Time period for sending integrated data to CH by CM

**Table 2 sensors-20-00913-t002:** Interval for each attribute used for data standardization.

	Time-Frequency Parameter	Residual Energy	Remaining Buffer Capacity	LQR	Distance to BS	Nsupport
Interval [min,max]	[0,100]	[0,2.5]	[0,100]	[10,15]	[150,10]	[0,40]

**Table 3 sensors-20-00913-t003:** Measured value after screening matrix.

	Time-Frequency Parameter	Residual Energy	Remaining Buffer Capacity	LQR	Distance to BS	Nsupport
S1	{70,65,60}	{2,1.8,1.6}	{78,75,73}	{11.5,11,10.5}	{110}	{30,27,25}
S2	{65,60,55}	{1.9,1.85,1.83}	{65,63,62,60}	{12}	{50,49}	{29,28,27}
S3	{65,60}	{1.9,1.7,1.5,1.3}	{90,87,84,82}	{12.5,11.9}	{45}	{29,26,24}
S4	{54,51,48,45}	{2.2,2,1.8}	{80,78,77}	{13.5,13,12.8}	{42,40}	{33,30}
S5	{67,63,59,55}	{2.1,1.9}	{95,92,91}	{14,13.8,13.5}	{39,38,36}	{31.5,29}

**Table 4 sensors-20-00913-t004:** Hesitant fuzzy decision matrix.

	Time-Frequency Parameter	Residual Energy	Remaining Buffer Capacity	LQR	Distance to BS	Nsupport
S1	{0.7,0.65,0.6}	{0.8,0.72,0.64}	{0.78,0.75,0.73}	{0.3,0.2,0.1}	{0.29}	{0.75,0.69,0.63}
S2	{0.65,0.6,0.55}	{0.76,0.74,0.73}	{0.65,0.63,0.62,0.6}	{0.4}	{0.7,0.71}	{0.74,0.7,0.68}
S3	{0.65,0.6}	{0.76,0.68,0.6,0.52}	{0.9,0.87,0.84,0.82}	{0.5,0.38}	{0.75}	{0.74,0.65,0.6}
S4	{0.54,0.51,0.48,0.45}	{0.88,0.8,0.72}	{0.8,0.78,0.77}	{0.7,0.6,0.56}	{0.76,0.78}	{0.82,0.75}
S5	{0.67,0.63,0.59,0.55}	{0.84,0.76}	{0.95,0.92,0.91}	{0.8,0.76,0.7}	{0.79,0.8,0.81}	{0.79,0.74}

**Table 5 sensors-20-00913-t005:** Entropy matrix.

	Time-Frequency Parameter	Residual Energy	Remaining Buffer Capacity	LQR	Distance to BS	Nsupport
S1	0.912	0.809	0.747	0.643	0.942	0.858
S2	0.961	0.767	0.918	0.987	0.890	0.832
S3	0.959	0.897	0.325	0.991	0.918	0.895
S4	0.999	0.644	0.683	0.943	0.808	0.786
S5	0.937	0.763	0.276	0.747	0.644	0.815

**Table 6 sensors-20-00913-t006:** Entropy coefficient weight values.

**Ei**	0.8185	0.8925	0.8308	0.8105	0.6970
**Wi**	0.1909	0.1131	0.1779	0.1993	0.3187

**Table 7 sensors-20-00913-t007:** Comparison of measured attribute values with threshold attribute values for CH election decision.

	α_σ(1)_ > D_F1_	α_σ(5)_ < D_F5_	α_σ(6)_ > D_F6_	CH Election Set	CH Election Decision
S1	Yes	No	Yes	No	S5 will be CH as it has highest attributes values
S2	Yes	Yes	Yes	Yes	
S3	Yes	Yes	Yes	Yes	
S4	No	Yes	Yes	No	
S5	Yes	Yes	Yes	Yes	

**Table 8 sensors-20-00913-t008:** Simulation Table.

Parameters	Values
Network Area (M × M)	100 × 100, 150 × 150, 200 × 200, 250 × 250, 300 × 300, and 350 × 350
Number of sensor nodes (N)	150
$E_{n},\;E_{int}\;\mathrm{and}\;E_{adv}$	1.5 J, 2 J and 2.5 J
$R_{normal}$, $R_{int}$, $R_{adv}$	40%, 35%, 25%
MAC layer	IEEE 802.15.4
$E_{elec}$	50 × 10^−9^ J/bit
$\varepsilon_{fs}$	10 × 10^−12^ J/bit/m^2^
$\varepsilon_{mp}$	0.0013 × 10^−12^ J/bit/m^4^
Energy consumed during aggregation $E_{A}$	5 × 10^−9^ J/bit
Energy consumed by CH during data fusion	5 × 10^−12^ J/bit
(Data + Overhead) Packet size	4096 bits

**Table 9 sensors-20-00913-t009:** Percentage Improvement of hesitant fuzzy entropy-based opportunistic clustering and data fusion scheme (HFECS) in terms of stability period.

Schemes	Network Size					
	100 × 100	150 × 150	200 × 200	250 × 250	300 × 300	350 × 350
SEP-E	27.88	32.55	26.09	35	52.63	81.48
DEEC-E	62.31	98.82	110.91	161.29	314.29	512.5
EBCS	−1.86	−3.43	−3.33	12.5	34.88	58.06
TEEN	45.52	53.64	54.67	62	87.1	122.73

**Table 10 sensors-20-00913-t010:** Percentage Improvement of HFECS in terms of half-life time.

Schemes	Network Size					
	100 × 100	150 × 150	200 × 200	250 × 250	300 × 300	350 × 350
SEP-E	25	23.81	27.03	41.5	53.39	74.16
DEEC-E	2.23	20.93	43.29	66.4	101.11	150
EBCS	−5.17	−3.7	−2.08	1.96	5.23	5.44
TEEN	31.7	31.31	38.24	56.39	75.73	101.3

**Table 11 sensors-20-00913-t011:** Percentage improvement of HFECS in terms of average residual energy.

Schemes	Number of Rounds						
	500	1000	1500	2000	2500	3000	3500
SEP-E	−1.47	2.78	13.64	68.75	600	328.57	233.33
DEEC-E	1.52	8.82	0	−25	−33.33	−58.33	−66.67
EBCS	−2.9	−2.63	8.7	50	600	328.57	233.33
TEEN	−3.6	−5.13	8.7	50	75	200	100

**Table 12 sensors-20-00913-t012:** Percentage improvement of HFECS in terms of network lifetime (FND).

Schemes	Network Size					
	100 × 100	150 × 150	200 × 200	250 × 250	300 × 300	350 × 350
SEP-E	18.87	17.78	24.72	35	45	58.06
DEEC-E	42.64	57.06	76	79.31	88	120
EBCS	−5.5	−6.47	0.91	22.73	38.1	8.89
TEEN	26.85	41.96	58.57	54.29	93.33	99.22

**Table 13 sensors-20-00913-t013:** Percentage improvement of HFECS in terms of network lifetime (HND).

Schemes	Network Size					
	100 × 100	150 × 150	200 × 200	250 × 250	300 × 300	350 × 350
SEP-E	11.39	8.64	15.38	34.19	44.8	58.95
DEEC-E	−2.94	8.64	34.73	57.58	92.55	96.1
EBCS	−1.86	−7.72	−2.26	6.67	10.37	7.86
TEEN	19.46	18.91	28.57	52.38	64.55	77.65

**Table 14 sensors-20-00913-t014:** Percentage improvement of HFECS in terms of PDR.

Schemes	Number of Rounds						
	500	1000	1500	2000	2500	3000	3500
SEP-E	3.26	16.18	28.57	36.11	50	66.92	97
DEEC-E	5.56	17.91	40	48.48	50	43.75	71.43
EBCS	0	−2.47	−1.56	4.26	9.09	9.52	20
TEEN	3.26	25.4	40	53.13	61.43	87.09	99

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
