# Peer review of "Hesitant Fuzzy Entropy-Based Opportunistic Clustering and Data Fusion Algorithm for Heterogeneous Wireless Sensor Networks"

_sensors, 2020, doi:10.3390/s20030913_

Round 1
Reviewer 1 Report
Title/Abstract: The data fusion algorithm/scheme is mentioned both in the title and in the second sentence of the abstract, but not discussed or explained any more until page 5. All focus seems to be on HFECS part and as a reader you start wondering why.
Abstract, line 8-9: Why do you use initial capital letters in “Cluster Formation and Cluster Head Election”? Same goes for “After Cluster information…” in the following sentence and “Our Simulation…” a bit further down. I would rather see this approach for the abbreviations HFECS and WSN, where you do not capitalize the initial letters. This occurs throughout the paper.
Page 2, Line 1: Do you have any proof or references backing the statement that you “…need a topology architecture in which sensor nodes are organized in clusters.” To use the resources energy efficient? Are there really no alternative solutions?
Page 2, Paragraph 3: I would consider revising this paragraph. There are a lot of words but the entropy is rather low (pun intended) in the second half. Probably it would be fine just leaving the first two sentences and removing the rest.
Page 3, Section 1: Here you have some strange capitalizations going on again.
Page 3, Section 3: You do not need to refer to [21] in three consecutive sentences, just refer to it once and then keep referring to the same authors/work/publication/survey or however you like to mention it.
Page 5, eq. 1 and 2: Here there are some serious issues. From eq. (1) it follows that En, Eint and Eadv represent the energy of all the normal sensor nodes, all the intermediate sensor nodes and all the advanced sensor nodes, respectively. However, I cannot see why the relations between these entities are relevant. It is clearly the individual intermediate and advanced nodes that need to have greater energy resources to conduct their individual tasks. Further, in equation (2), the number of nodes (N) is included. Based on that, I cannot see how the two representations of Etotal (1) and (2) can be consistent. As an example, consider what happens when you substitute N for an arbitrary large number. As this number is increased, so is the value of (2) while (1) stays constant. I short, I think (2) makes sense whereas (1) and the reasoning in the sentences just before (1) do not. Also, in the last paragraph oh page 8, it seems like En, Eint and Eadv represents the energy of each individual type of node, not all of them. This seems much more reasonable.
Page 6: There should not be indentations in the text directly after the equations unless there are new paragraphs. In general, on page 6 this is not the case, as there are not even new sentences. This appears on other pages as well.
Page 6, eq. 3. Introduce different notations for free-space and multi-path fading cases as well while you are at it.
Page 6, Paragraph 3: Regarding the sensing, how do you know when (and how long) to sense with this asynchronous scheme?
Page 6, eq. 5-6: I see no reason why this should not be made into a single equation containing two equality symbols (=). This will also make it clearer that you are performing substitutions into the equation, not presenting two different definitions of the same notation.
Page 6, eq. 7-8: See my previous comment.
Page 6, Last paragraph: The statement of number of nodes per cluster must be in average. Clarify this.
Page 7: Consider swapping 3.2 and 3.1. I think this would make more sense. As an example, the reader will get to see your reasoning regarding position awareness and the use of RSSI before seeing eq. (3).
Page 7, fig 1: The green colors are very alike (i.e. CH and advanced sensor node), change one! Let the arrowheads end right before the nodes instead of overlapping them, some CHs look more red than green. Import the image in a different format such that the lines get sharp, they are fuzzy at the moment.
Page 8, Last paragraph: As I have already written, this use of En, Eint and Eadv makes more sense. However, I recommend you to rephrase “…uneven energy consumption…” as the consumption is considered to be even by your own definition! it is the consumed energy in relation to the available energy that is uneven.
Page 9, eq. 11-12: Why bother with defining CA? As far as I can see you do not use it anywhere else and it is mostly a waste of space.
Page 10, Alg. 1: This can be compressed a bit, e.g. refer to eq. (12) and (13) instead of rewriting them. Why do you need to separate the cases with multiple or single sensor nodes with max value (Line 24-28)?
Page 11, Alg. 2: On line 10, it seems strange to use the word “carefully” in an algorithm description.
Page 11, Paragraph 2: The first sentence is very long (it covers 5 rows). It needs to be divided. Also, when multiple announce messages are received, is it not possible that more than one of them signals attribute values >DF4?
Page 12, Line 1: How should the equation starting with ”j=1:N/Nc-1, TF…” be interpreted?
Page 12, Alg. 3: On line 11-12 I suggest that you simply refer to eq 15 instead of repeating it. Alternatively, use the equations/notations and reduce the amount of text. This comment is valid for most of the algorithm descriptions.
Page 13, middle: How can i=j and at the same time be different cluster members?
Page 15, Section 5: You write “In this paper, we have demonstrated the opportunistic clustering mechanism…”. I have not seen any demonstrations, only a description of it.
Page 16, Fig. 2: text almost unreadable due to the fuzziness. Import in a proper format!
Page 18, Table 7: Please motivate why these are relevant/realistic numbers.
Page 19, 6.2: Choose if you like to present the geographical network size here or in 6.1 on the previous page. Now it is presented in both subsections, which is redundant.
Page 21-, 6.2.3: I can see the value and relevance of pesenting the stability period and half-life time, but why is the residual energy of such interest that it has been devoted 3 different graphs (fig. 5-7)?
Page 23, last paragraph: Here you introduce FND and HND. To me that seems to be the exact same thing as you previously called stability period and half-life time.
Page 24, Fig 8: I find this graph hard to read, and see no reason presenting it as the information is already available in fig 3 and 4, which are much easier to read. Or am I missing something? The same question is valid for Fig. 9, although I can see some minor differences when compared to Fig. 3 and 4. Please explain!
Page 25, Fig 9: The image quality of this graph needs to be improved, it is for some reason much worse than the quality of the rest of the graphs.
Page 25-26, Table 11-13. I suggest that you settle for a two-figure representation of the results. I am quite convinced that the generalizations and assumptions you make in the simulations give rise to greater deviations than tens or hundreds of a percent.
Final, general, comments:
This is an interesting paper, but there are some issues that need to be addressed. My main concern is regarding eq. (1) and (2), see previous comments, and if there might be similar problems with other (more complex) equations that I have not been able to detect.
A list of notations would have been nice (as there are many of them). As a reader you cannot manage to remember them all and it is tedious to search the paper to find where they are introduced. This will of course require some extra space, but this can handled by removing some of the redundant figures and shortening the algorithm presentations. The main text can probably also benefit from being condensed somewhat.
A proof reading would be beneficial. The English grammar is mostly ok, but not perfect. In some places, the “s” is missing at the end of the verb after singular nouns, and sometimes the/a/an is missing when they should be present in front of the nouns. Finally, some sentences are too long and should be divided.
Reviewer 2 Report
The paper clearly gives the environment, assumptions, requirements and objectives of the problem in hand, and points out major issues or difficulties when dealing with the problem and the system design. However, it can be improved as follows:
(1) My main concern with this manuscript is that the proposed scheme can be explained better since the main description may not be very clear for the readers. The organization, presentation, performance evaluation of the paper can be improved.
(2) What is the impact of the network topology on the system performance? For instance: consider different sensor deployment strategies: (I) making a random uniform deployment, (II) making sensor density high at the center of the terrain, and (III) making a checkerboard grid deployment. It would be better to have a discussion about this issue.
(3) Spell check is required.
Reviewer 3 Report
The authors claim to use hesitant fuzzy entropy to form heterogeneous WSN. The description of algorithms and analyses on details are presented sufficiently well.
However, it is not very clearly explained why this new measure of hesitant fuzzy entropy needs to be used for WSN formation. In other words, what point of the proposed algorithm makes the proposed algorithm competitive with existing methods?
According to the simulation results, there are inversions in performance with other algorithms like EBCS. This means that the proposed algorithm is not intended to directly handle specific objectives that are presented in simulation sections. Therefore, it is suggested to state clearly, in introduction or system model section, what features of the proposed strategy makes the authors working on it. Also, the reasons of performance inversions are clearly discussed in all figures pertaining to simulation results
Reviewer 4 Report
The scheme presented provides original contribution. However, some parts of the paper are of concern and should be clarified. See my comments below.
The authors’ simplified energy model assumes that transmission power and hence energy consumed in transmission depends on distance squared or the fourth power of the distance. Although it is a reasonable assumption that that the required power and energy fits this model, it is not reasonable to assume that a node can tune its power so precisely. (Also, epsilon_fs and epsilon_mp are not the energy dissipated. The energy is the entire expression that multiplies epsilon.) The authors state that “Every sensor node has two adjustable radio ranges i.e. short radio range ?? and long radio range ??. ” This statement implies that the nodes have only two transmission power settings, which is inconsistent with equation 3. Equation 3 indicates that transmission energy would depend precisely on the transmission distance. Although nodes can estimate distances using RSSI, it is not reasonable to assume they can precisely tune their transmission power based on distance. The authors do not state if two transmission powers were used in simulations, or something else.
The authors state in their conclusion that “Our simulation results clearly show that HFECS performs better than existing benchmarks.” However, their results show that EBCS (Figures 3 and 4) or DEECE-E (Figures 5 and 6) give better results for small network size. The authors should state clearly under what conditions their scheme outperforms others.
Specific comments on the text:
-Table 1 is located before its reference in the text, whereas Table 2 is located after it is referenced in the text.
- The quality of the text in the rectangle of Figure 2 is poor.
Numerous sentences omit articles (the, a), which makes them difficult to read. Although in some cases it may be a matter of style, adding articles would improve readability in my view. Here are my comments on English:
Section 5 describes the case study related to the HFE based clustering scheme in neighborhood area networks.
Performance evaluation and simulation results can be depicted are described in Section 6.
Finally, Sections 7 and 8 discusses and concludes the paper respectively, and provide some future research directions as well.
...but no clustering scheme considers load on the cluster in terms of sensor nodes in its design except ‘Entropy-based Clustering Scheme (EBCS)’ in [39].
Furthermore, the entropy weight coefficient technique is used for optimal CH selection in every round.
…the role of CHs within a cluster needs to be substituted using some decision criteria in order to avoid hotspot problem.
As Since we are dealing with MADM, so we have considered several parameters like time frequency…
In the HFE based clustering scheme, the BS executes the Election procedure and creates a CH Election set.
that neighboring sensor node but if the CH_announce message is sent by a single sensor node, then CM send Con
which receives the aggregated data from multiple NAN gateways and forwarding it to the control center [46].
…the standardized values are {([150,136]→[0,0.1]),([136,122]→[0.1,0.2]),……([66,52]→[0.6,0.5])…([24,10]→[0.9,1.0])} due to the reason that because if ‘Distance to BS’ is minimum, then it will lead towards better connectivity with the BS.
Moreover, we also find the average data value in the measured value matrix and then we synthesize the final data values of all measured attributes using Equation (14).
In addition to it, EBCS achieves better half-life time as compared to HFECS for network size up to 200x200 m2 due to key differences between EBCS and HFECS.
Round 2
Reviewer 2 Report
The authors have satisfactorily addressed all of the main concerns.
Reviewer 3 Report
Authors have properly addressed the comments raised by the reviewer.